# Safety and Humoral and Cellular Immunogenicity of the BNT162b2 SARS-CoV-2 Vaccine in Liver-Transplanted Adolescents Compared to Healthy Adolescents

**DOI:** 10.3390/vaccines10081324

**Published:** 2022-08-16

**Authors:** Palittiya Sintusek, Supranee Buranapraditkun, Siriporn Khunsri, Varattaya Saengchaisukhonkit, Preeyaporn Vichaiwattana, Donchida Srimuan, Thanunrat Thongmee, Yong Poovorawan

**Affiliations:** 1Thai Pediatric Gastroenterology, Hepatology and Immunology (TPGHAI) Research Unit, Division of Gastroenterology and Hepatology, Department of Pediatrics, King Chulalongkorn Memorial Hospital and Thai Red Cross Society, Faculty of Medicine, Chulalongkorn University, Bangkok 10330, Thailand; 2Center of Excellence in Vaccine Research and Development (Chula VRC), Division of Allergy and Clinical Immunology, Department of Medicine, Chulalongkorn University, Bangkok 10330, Thailand; 3Center of Excellence in Clinical Virology, Faculty of Medicine, Chulalongkorn University, Bangkok 10330, Thailand

**Keywords:** liver transplant, BNT162b2, adolescents, immune response, vaccine

## Abstract

Since BNT162b2 was approved to prevent COVID-19 in children, we aim to compare the safety and immunogenicity of the BNT162b2 vaccine in liver-transplanted (LT) and healthy adolescents. LT and healthy adolescents received two doses of 30 µg of BNT162b2. All were evaluated for total COVID-19 antibodies directed against the receptor-binding domain (RBD) and interferon-γ using the ELISpot at all time points; anti-nucleocapsid immunoglobulin was evaluated at week 8 and the surrogate virus-neutralizing antibody (sVN) to Omicron at day 0 and week 8. Adverse effects were recorded during days 0–7. In total, 16 LT and 27 healthy adolescents were enrolled (aged 14.78 ± 1.70 years). After completion, all LT and healthy adolescents were positive for anti-RBD immunoglobulin, with geometric mean titers of 1511.37 (95% CI 720.22–3171.59) and 6311.90 (95% CI 4955.46–8039.64)) U/mL (*p* < 0.001). All tested negative for anti-nucleocapsid immunoglobulin, indicating no COVID-19 infection after vaccination. However, the sVNs to Omicron were positive in only nine (33.33%) healthy adolescents and none of the LT adolescents. Interferon-γ-secreting cells were lower in LT adolescents than healthy adolescents. The LT adolescents had a lower immunogenic response to BNT162b2 than the healthy adolescents. Administrating two doses of BNT162b2 was safe, but was less effective against the Omicron variant.

## 1. Introduction

Severe acute respiratory syndrome coronavirus 2 (SARS-CoV-2) emerged for the first time in Wuhan, China, in December 2019 [1]. This infectious disease spread globally and was declared by the World Health Organization (WHO) as a pandemic on 11 March 2020 [2]. More than 579 million people were infected with SARS-CoV-2, and unfortunately, more than 6.4 million deaths occurred [3]. Vaccines are considered the most powerful strategies to not only prevent disease, but also diminish the severity of the disease.

Numerous vaccine candidates have been developed, and a few were approved for use by the authorities, including an mRNA vaccine, BNT162b2. Due to its high efficacy and favorable safety profile, the BNT162b2 vaccine was authorized for use in adults by the Food and Drug Administration (FDA) in December 2020, a year after the start of the pandemic [4]. Since BNT162b2 was approved to prevent SARS-CoV-2 in children > 12 years old in May 2021 [5], Thailand created a policy to provide BNT162b2 to adolescents, primarily immunocompromised adolescents, in August 2021.

Although there has been strong evidence that liver transplantation does not confer major additional susceptibility to adverse outcomes from SARS-CoV-2, the benefit of immunization for this vulnerable population is still accepted [6,7]. A return to school for education and social development is indeed needed for both healthy and liver-transplanted children [8]. However, the safety of vaccines for SARS-CoV-2 should be monitored carefully. To date, there are limited data on the humoral immune response to BNT162b2 and no data on the cellular immune response to BNT162b2 in liver-transplanted children. Hence, the aim of the present study was to compare the safety and immunogenicity of the BNT162b2 vaccine in liver-transplanted adolescents and healthy adolescents. The factors that might affect the immune response to SARS-CoV-2 vaccines were evaluated.

## 2. Methods

### 2.1. Study Design and Target Population

The present study prospectively evaluated the immunogenicity of SARS-CoV-2 vaccination with BNT162b2 in liver-transplanted adolescents who had regularly follow-ups at King Chulalongkorn Memorial Hospital (KCMH), Bangkok, Thailand, and healthy adolescents aged between 12 and 17 years old. The exclusion criteria for liver-transplanted children included participants who underwent liver transplant within the last 6 months, had a history of previous or active SARS-CoV-2 infection, and were unstable at the time of vaccination with a fever higher than 38 degrees Celsius. The exclusion criterion for healthy adolescents was a history of previous or active SARS-CoV-2 infection.

This clinical trial was registered in the Thai Clinical Trial Registry (TCTR20210830002) and approved by the Institute Research Board at Chulalongkorn University, Thailand (IRB No 715/64).

After completion of informed consent, participants received 30 µg of BNT162b2 injection on day 0 and days 21–28. Blood samples were collected before vaccine administration at 4 time points: prior to the first dose, days 21–28, week 5 and week 8 after the first dose. In liver-transplanted adolescents, the baseline characteristics, basic laboratory results and history of immunosuppressant use prior to immunization were collected for analysis. In healthy adolescents or the control group, the baseline characteristics, including age, sex, body weight, height and vital signs, were collected for analysis.

### 2.2. Adverse Effects

All participants were observed in the clinic for 30 min after vaccination for immediate adverse effects. Then, specific local and systemic symptoms after vaccination were recorded via a Google form on days 0–7 after the first and second doses of BNT162b2. The liver-transplanted adolescents were monitored at the transplant clinic for liver injury up to 6 months after the first dose of BNT162b2 administration.

### 2.3. Immunogenicity Study

All serum samples were evaluated for total immunoglobulins (Igs) specific to the receptor-binding domain (RBD) of the SARS-CoV-2-spike (S) protein using an Elecsys SARS-CoV-2 S electrochemiluminescence immunoassay (ECLIA) according to the manufacturer’s instructions (Roche Diagnostics, Basel, Switzerland) as previously described [9]. Anti-RBD Ig levels less than 0.8 U/mL were considered negative.

Anti-nucleocapsid (anti-N) IgG was tested for in a subset of serum samples (week 8) by using a chemiluminescent microparticle immunoassay (CMIA) according to the manufacturer’s instructions (Abbott Diagnostics, Sligo, Ireland).

Neutralizing antibodies (NAs) against Omicron variants were assessed in a subset of serum samples (day 0 and week 8) by using a surrogate virus-neutralization test (sVNT) based on an antibody-mediated blockade of the interaction between the viral receptor-binding domain (RBD) and angiotensin-converting enzyme 2 (ACE2) protein. The cPass SARS-CoV-2 neutralization antibody detection kit was used (Nanjing GenScript Biotech Co., Ltd., Nanjing, China) as previously described [9]. The percent inhibition was calculated as inhibition (%) = (1: OD value of sample/average OD of negative control) × 100. A value ≥ 30% scored positive, indicating the presence of neutralizing antibodies.

The number of interferon-γ-secreting T cells were evaluated in PBMCs using a interferon-γ enzyme-linked immunospot (ELISpot) assay according to the manufacturer’s instructions (Mabtech, Stockholm, Sweden) as previously described [10]. Results are expressed as spot-forming cells (SFCs/10^6^ PBMCs) after subtracting background spots from the negative wells.

### 2.4. Vitamin D Study

Quantitative determination of the total 25OH vitamin D in all serum samples was assessed by using a direct competitive chemiluminescence immunoassay (CLIA) according to the manufacturer’s instructions (LIAISON 25OH vitamin D assay, DiaSorin, MN, USA).

### 2.5. Statistical Analysis

Continuous and categorical variables were expressed as the median (interquartile range or IQR) or mean (SD) and by percentage or count as appropriate. The comparison of categorical variables was performed using Fisher’s exact test. Continuous data were compared using Student’s *t* test or the Mann-Whitney U test as appropriate. Geometric mean antibody titers were calculated for the antibody value and are presented with a 95% confidence interval (CI). A logistic regression analysis model was used to assess the potential factors associated with the immune response after vaccination. A *p* value < 0.05 was considered statistically significant. Statistical analysis was performed using STATA version 15 (StataCorp, College Station, TX, USA). Graphs and figures of the analyzed data were generated using GraphPad Prism version 9.2.0 for Windows (GraphPad Inc., San Diego, CA, USA).

## 3. Results

### 3.1. Baseline Characteristics of the Liver-Transplanted and Healthy Adolescents

There were 32 liver-transplanted adolescents aged between 12 and 17 years who had regular follow-ups at KCMH. Three of them received the first dose of the ChAdOx1-S vaccine, one underwent liver transplantation 3 months before the vaccination and eleven of them could not be vaccinated at KCMH and decided to receive the vaccine at a nearby hospital. One participant had positive anti-RBD immunoglobulin before vaccination that reflected a previous asymptomatic COVID-19 infection. Hence, 16 liver-transplanted adolescents were recruited in the present study. The mean (SD) age was 14.59 (1.78) years (36.8%, male). Comorbidities included splenectomy (*n* = 2), anticoagulant use (*n* = 2), chronic hepatitis E infection (*n* = 1), bile duct stricture (*n* = 1), suspected primary sclerosing cholangitis (PSC) (*n* = 1), hemoglobin H disease (*n* = 1), attention deficit hyperactivity disorder (ADHD) on Ritalin (*n* = 1), polycystic ovarian syndrome (PCOS) (*n* = 1), unidentified cause of chronic hepatitis (*n* = 1) and ascending sclerosing cholangitis (ASC) with inactive ulcerative colitis (UC) (*n* = 1). The calcineurin inhibitor (CNI) was the major immunosuppressant (10 tacrolimus and 6 cyclosporin). CNI monotherapy was given to 10 patients; 4 were given two immunosuppressants (tacrolimus and azathioprine to 1 patient; tacrolimus and mycophenolate mofetil (MMF) to 3 patients). Two patients received three immunosuppressants because of recurrent ASC and UC in one case (tacrolimus + MMF + prednisolone) and a chronic unidentified cause of hepatitis in one case (tacrolimus + sirolimus + prednisolone). The baseline characteristics and immunogenicity after BNT162b2 treatment are shown in Table 1.

### 3.2. Immunogenicity of the BNT162b2 Vaccine

#### 3.2.1. Humoral Response

All participants in the liver transplant group and healthy control group were positive for anti-RBD immunoglobulin after the second dose of the BNT162b2 vaccine. There was significantly lower anti-RBD immunoglobulin in the liver-transplanted adolescents than in the healthy adolescents at all visits (*p* < 0.05) (Figure 1, Table 1). Moreover, all were negative for anti-N IgG at visit 4, reflecting no SARS-CoV-2 infection during the study period which had a mean time of 56.28 (1.80) days (data not shown). However, there was a lower percentage of inhibition of sVNT to the Omicron variant in the liver-transplanted adolescents compared to the healthy adolescents (8.23 (2.35, 16.73) vs. 15.36 (4.61, 19.44) %, *p* = 0.066). According to the cut-off value of % sVNT inhibition to SARS-CoV-2, none of the liver-transplanted and only nine (33.33%) of the healthy adolescents had positive results (>30% inhibition) (Figure 2, Table 1).

#### 3.2.2. Cellular Response

The T-cell-specific response to SARS-CoV-2 spike peptide pools (S1 and S2) was assessed using an interferon-γELISpot assay at all time points. There was a statistically significant increase in the number of interferon-γ-secreting cells after the first and second dose of BNT162b2 at week 3 and week 5 (*p* = 0.0192 and *p* = 0.0004, respectively), but no statistically significant increase at week 8 (*p* = 0.247) (Figure 3, Table 1).

### 3.3. Factors Associated with the Immunologic Response to BNT162b2 Administration in Liver-Transplanted Adolescents

Regarding the potential factors associated with the immunologic response, we divided the LT adolescents into low (*n* = 4) and high (*n* = 12) anti-RBD immunoglobulin levels using the cut-off value of 550 U/mL after two doses of BNT162b2 (the value of the 25th percentile of anti-RBD immunoglobulin from liver-transplanted adolescents) as compared to each factor (Table 2). A history of graft rejection, prednisolone use, number of immunosuppressants used, SGOT, SGPT and % sVNT inhibition to the Omicron variant reached statistical significance from univariate analysis. Moreover, there was a strong correlation between anti-RBD immunoglobulin and % sVNT inhibition for the Omicron variant (*r* = 0.7351, *p* < 0.001) (Figure 4). Hence, correlation analysis was further assessed for the % sVNT inhibition of the Omicron variant with SGOT and SGPT. SGOT and SGPT had a Pearson’s correlation of −0.3702 (*p* = 0.221) and −0.3238 (*p* = 0.158), respectively.

Other significant factors associated with the immunologic response from the present study and previous studies were further analyzed using logistic regression. After adjustment for potential confounders (age, sex, body mass index), the duration from liver transplantation to vaccination and combined immunosuppressant therapy were the significant factors associated with the % inhibition of sVNT to the Omicron variant (*p* < 0.05) (Table 3).

### 3.4. Safety of BNT162b2

Most of the participants in the liver transplant group and healthy control group tolerated the first and second doses of the BNT162b2 vaccine 7 days after administration. The most common local adverse effects were pain in the arm, swelling at the injection site and redness. The most common systemic adverse effects were myalgia, headache, joint pain and fever. There were no serious adverse events or adverse effects that required hospitalization in either group. There was no statistically significant difference in adverse effects between the liver transplant and healthy control groups (Figure 5).

## 4. Discussion

A two-dose regimen of 30 µg of BNT162b2 elicited a high humoral immune response in liver-transplanted adolescents and healthy adolescents in the present study. All healthy adolescents and 93.8% of liver-transplanted adolescents had immune responses after the first dose. Furthermore, all participants had an immune response after the second dose. However, the GMT of the anti-RBD and T-cell-specific response to SARS-CoV-2 spike peptide pools (S1 and S2) in liver-transplanted adolescents was significantly lower than that in healthy adolescents within 2 months after the first dose. Moreover, none of the liver-transplanted adolescents and only 33.33% of healthy adolescents had a positive sVN antibody to the Omicron variant. Regarding the short-term adverse effects of the BNT162b2 vaccine, there were minimal local and systemic adverse events in both groups, but the difference was not statistically significant. The potential factors of an attenuated immune response were the time from liver transplantation to vaccination and the amount of immunosuppressants used.

To our knowledge, this is the first study to demonstrate both humoral and cellular immune response and safety of BNT162b2 in liver-transplanted children. Recently, published reports and studies in liver-transplanted adults demonstrated that the immune response to BNT162bs was 47.5% [11], 72% [12] and 79% [13]. The response rate varies depending on age, combined immunosuppression, type of immunosuppression (MMF), time from liver transplantation to vaccination and comorbidities (renal impairment and obesity) [11,12,13,14]. Additionally, there has been only one published study of the antibody response to a second dose of BNT162b2 in solid-organ-transplanted adolescents; that study, conducted by Qin CX et al., included 25 (44%) liver transplant recipients. They found a seropositive rate of 89.47% (17/19) after the second dose of BNT162b2, and the factors associated with poor immune response were time from liver transplant (<3 years), multiple immunosuppressants and use of antimetabolite drugs [15]. The result of the present study is in line with the study by Qin CX et al. Liver-transplanted adolescents had a higher antibody response to BNT162b2 than liver-transplanted adults, but a lower response than healthy adolescents. However, with the emergence and spreading of the Omicron variant, there are increasing concerns about the adequate protection of two doses of BNT162b2 in the population. Therefore, we further evaluated the sVNT antibody specific to the Omicron variant and found that liver-transplanted and healthy adolescents had a poor immune response to this broader array of mutations than prior variants (B.1.6.17.2/Delta variant). Even though no participants had COVID-19 during the follow-up period, our results strongly demonstrated the lower effectiveness of two doses of BNT162b2 for the Omicron (B.1.1.529) strain. Hence, an additional booster dose and further study of both the antibody response and real-world protection are needed. In addition, other types of neutralizing antibodies that are more specific, such as a pseudovirus neutralization assay for the Omicron strain, might be evaluated to confirm the ineffectiveness of BNT162b2 to the Omicron variant as found in this study. Regarding the role of cellular response to COVID-19 infection, the virus has the ability to spread from cell to cell without exposure to the extracellular environment [16], which might explain why some people who have high antibodies were infected with COVID-19. T helper 1 cells are considered to have a pivotal role in the host responses to COVID-19 infection, reducing viral spreading and also supporting B cell function [17,18]. Hence, robust cellular immunity could promote the long-term protection and reduce the risk of severe disease [19]. In the era of frequent COVID-19 mutation, the available vaccines right now seem to be less effective for disease prevention, but can decrease the disease severity. The present study demonstrated the rigorous response of T cells against the S protein in both liver-transplanted adolescents and healthy adolescents; the association of IFN-γ-secreting T cells with disease prevention and disease severity in infected participants deserved to be studied in our ongoing research.

Recent data about the burden of COVID-19 infection in liver-transplanted children remain conflicting. The data from 10 European centers demonstrated the more frequent hospitalization and complications of liver-transplanted children compared to children with chronic liver diseases who were infected with COVID-19 [20], whereas the NASPGHAN/SPLIT SARS-CoV-2 International Registry found that liver-transplanted recipients were less likely to require hospitalization and intensive care units compared to children with liver diseases [21]. The different etiologies of liver diseases between the two studies might explain this result, as obesity and nonalcoholic fatty liver disease (NAFLD), strong factors of severe COVID-19 manifestations, were prominent in the chronic liver disease group in the NASPGHAN/SPLIT SARS-CoV-2 International Registry. However, the cost-effectiveness of vaccination to prevent infection should be weighed with the safety of this vaccine in liver-transplanted children. Although the safety of BNT162b2 was evidenced in healthy adolescents [22], data on liver-transplanted children are scarce. Although this was a small study with a short-term follow-up period, the common adverse effects of the vaccine were mainly local reactions and myalgia, and there were no serious adverse effects. Data from a larger population are also needed. In our opinion, the safety of the vaccine should be monitored longer, and the suspected serious adverse effects should be vigorously reported. The safety of each vaccine used in liver-transplanted children should be evaluated in parallel with the efficacy. Previous studies have reported strong immune dysregulation after mRNA vaccination, including autoimmune hepatitis, idiopathic thrombocytopenic purpura, and acute disseminated encephalomyelitis (ADEM) [23,24]. Hence, liver-transplanted children who had transaminitis of an unknown cause or de novo hepatitis after liver transplantation should avoid the mRNA vaccine and choose an alternative vaccine if possible.

The present study found that the time from liver transplant to vaccination and combined immunosuppressants were the factors associated with a low immune response. However, MMF used in the present study was not the crucial factor that affected the immune response after BNT162b2 vaccination as in previous studies [11,13]. All of the liver-transplanted adolescents had an immune response after BNT162b2 vaccination, and the small number of participants with MMF used in the present study might explain why the result did not reach statistical significance.

There is strong evidence that vitamin D enhances both innate and adaptive immunity and benefits patients who are infected with COVID-19. Low vitamin D levels are an independent risk factor for COVID-19 infection, morbidity and mortality according to many observational studies [25,26,27,28]. Furthermore, a more ideal vitamin D status could improve the immune response to influenza vaccination [28]. Hence, there is a hypothesis that vitamin D supplementation can also promote the production of antibodies from B cells that are dependent on T cells after the COVID-19 vaccine [29]. The present study is the first to investigate total 25OH vitamin D levels in all participants before BNT162b2 immunization. We found that liver-transplanted adolescents had higher total 25OH vitamin D levels than healthy adolescents and higher total 25OH vitamin D levels in the subgroup of liver-transplanted adolescents who had anti-RBD IgG > 550 U/L. Because of our hospital protocol, we monitored vitamin D levels and provided adequate supplementation in liver-transplanted adolescents who had low vitamin D status. The higher level of vitamin D in liver-transplanted adolescents but the lower immune response after BNT162b2 might explain the independent factors mentioned previously. When comparing liver-transplanted adolescents who had high and low immune responses after BNT162bs treatment, vitamin D levels tended to be significantly higher in the group with a high immune response. The better vitamin D status may contribute to the effectiveness of BNT162b2 immunization. This speculation warrants further study in a larger population.

The limitation of the present study was the low number of participants from a single center. The difference in the timing of the second vaccination between liver-transplanted adolescents and healthy adolescents may affect the immune response to the BNT162b2 vaccine. In addition, the present study did not include long-term follow-up regarding the persistent immune response. However, the strength of the present study was that it studied the immunogenicity of BNT162b2 in both the humoral and cellular response. The IFN-gamma cytokines were evaluated, which are important cytokines that secret cells from both CD4+ and CD8+ effector memory T cells and are considered the indicator for immune response after vaccine immunization. However, other cellular responses that are also valuable to investigate in further studies include the TNF-α-secreting or IL-2-secreting CD4+ and CD8+ T cells. 

Last, but not least, the present study also provided up-to-date data regarding the ineffectiveness of two doses of BNT162b2 with regard to the Omicron variant. Hence, strategies to prevent infection from this mutation deserve further study. Today, the pre-exposure prophylaxis of COVID-19 using monoclonal antibodies is very promising, especially in these vulnerable patients who may not mount an adequate immune response to COVID-19 vaccination or have a severe adverse reaction to the COVID-19 vaccine [30]. Moreover, in infected cases, strategies to treat infection including antiviral agents, immunotherapy and passive immunity are also crucial and are currently being tested in randomized clinical trials [31].

## Figures and Tables

**Figure 1 vaccines-10-01324-f001:**
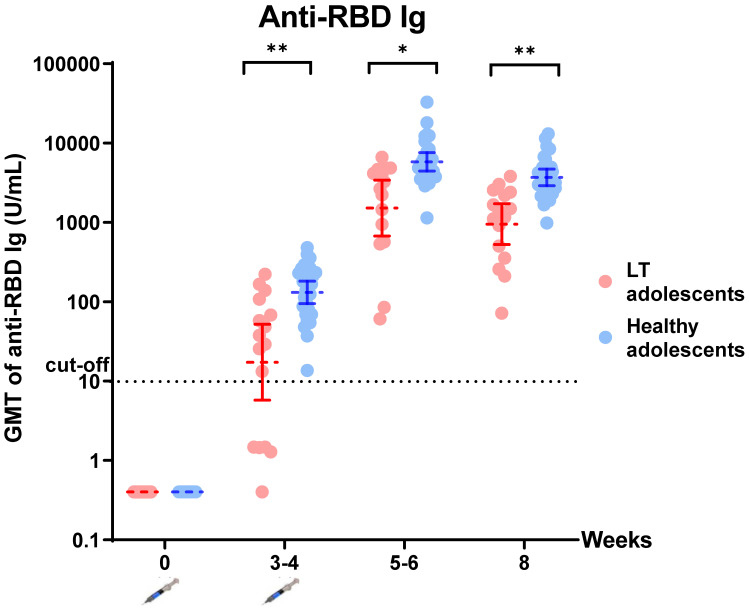
Comparing anti-RBD immunoglobulin (Ig) after completion of BNT162b2 vaccination in liver-transplanted adolescents and healthy adolescents. Data points are reciprocals of the individual. Line indicates geometric mean titer and bar indicates 95% confidence interval. * indicates *p* < 0.05; ** indicates *p* < 0.01.

**Figure 2 vaccines-10-01324-f002:**
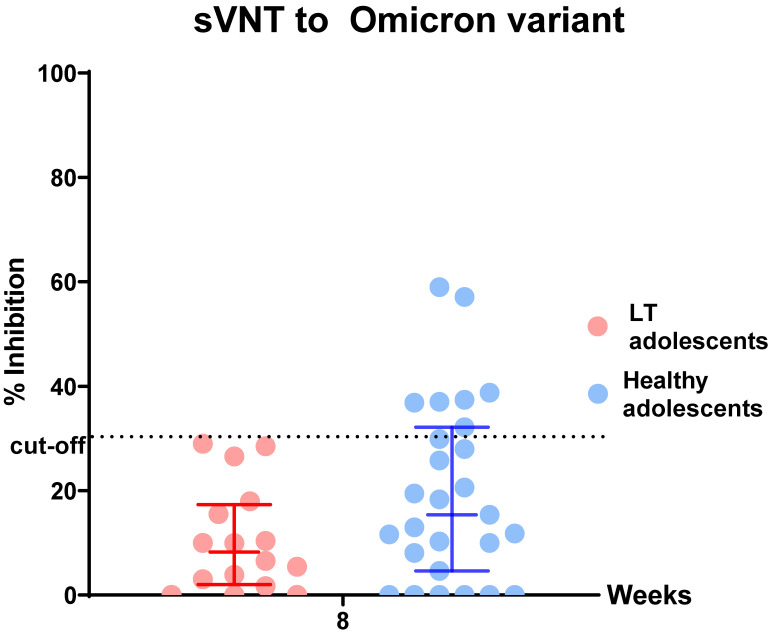
Comparison of % sVNT inhibition to the Omicron variant after completion of BNT162b2 vaccination in liver-transplanted adolescents and healthy adolescents. Data points are reciprocals of the individual. Line indicates geometric mean titer and bar indicates 95% confidence interval.

**Figure 3 vaccines-10-01324-f003:**
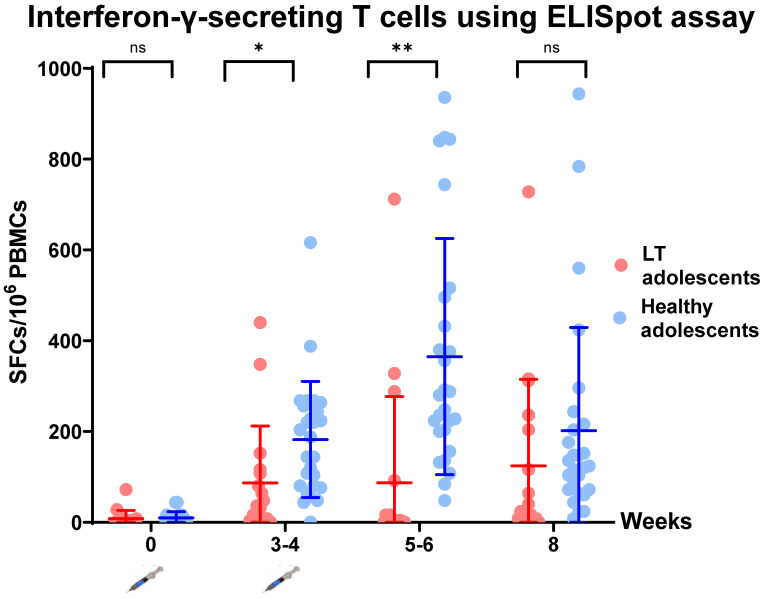
Comparison of T-cell-specific response to SARS-CoV-2 spike peptide pools (S1 + S2) measured using interferon-gamma ELISpot. Data points are reciprocals of the individual. Line indicates geometric mean titer and bar indicates 95% confidence interval. * indicates *p* < 0.05; ** indicates *p* < 0.01. ns—no statistical significance.

**Figure 4 vaccines-10-01324-f004:**
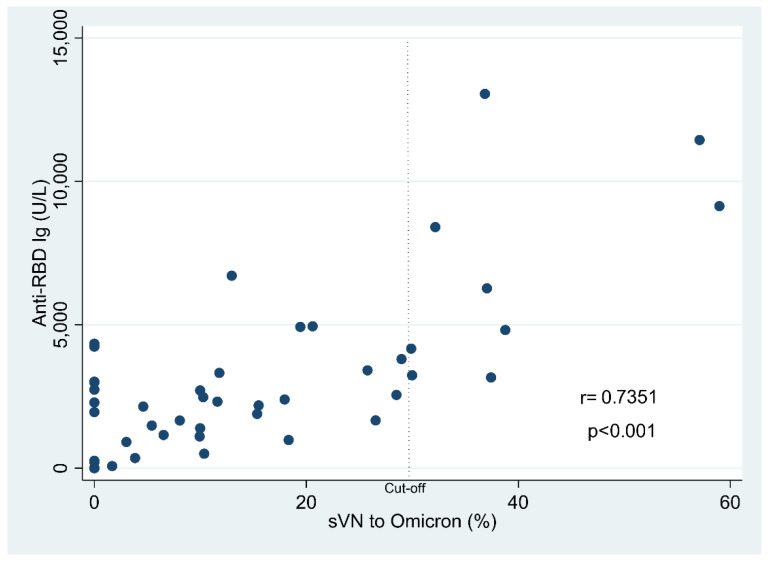
The Pearson’s correlation between anti-RBD immunoglobulin and % inhibition of sVNT to Omicron variant.

**Figure 5 vaccines-10-01324-f005:**
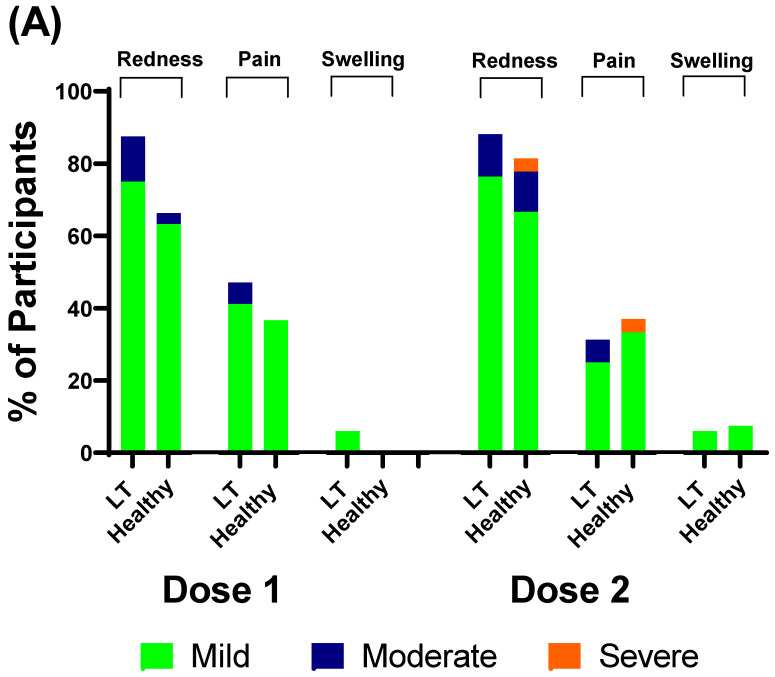
Adverse events during the 7 days after the first and second doses of BNT162b2 administration. (**A**) Local adverse events; (**B**) systemic adverse events.

**Table 1 vaccines-10-01324-t001:** Participant characteristics and immunologic response to BNT162b2 vaccination between liver-transplanted adolescents and healthy adolescents.

Parameter	Liver-Transplanted Adolescents (*n* = 16)	Healthy Adolescents (*n* = 27)	*p* Value
Age, years (mean ± SD)	14.59 (1.78)	14.67 (1.57)	0.441
Male gender, *n* (%)	7 (43.8)	12 (44.4)	0.609
Body mass index (kg/m^2^) (mean ± SD)	18.59 (2.52)	19.62 (4.27)	0.399
Underlying disease for LT, *n* (%)		-	-
Biliary atresia	9 (56.25)		
Budd–Chiari syndrome	1 (6.25)		
Alagille syndrome	2 (12.5)		
Biliary hypoplasia	1 (6.25)		
Citrullinemia	1 (6.25)		
Autoimmune hepatitis	1 (6.25)		
Cryptogenic cirrhosis	1 (6.25)		
Time from liver transplant to BNT162b2 vaccination (years) (mean ± SD)	7.18 (5.12)	-	-
History of graft rejection, *n* (%)	4 (25)	-	-
Type of immunosuppressant, *n* (%)		-	-
Tacrolimus (calcineurin inhibitor)	10 (62.5)		
Cyclosporin (calcineurin inhibitor)	6 (37.5)		
Sirolimus (mTOR)	1 (6.25)		
MMF (antimetabolite)	4 (25)		
Prednisolone	2 (12.5)		
Azathioprine	1 (6.25)		
Immunosuppressant used (*n*), *n* (%)			
1	10 (62.5)		
2	4 (25)		
3	2 (12.5)		
Basic laboratory data (mean ± SD)		-	
Hb (g/L)	135.12 (22.31)		
WBC (10^3^/µL)	5.38 (1.69)		
Absolute neutrophil counts	2830 (910)		
Absolute lymphocyte counts	2000 (890)		
Platelet (10^3^/microL)	203.94 (60.86)		
Total bilirubin (µmol/L)	21.38 (17.96)		
SGOT (U/L)	47.94 (47.74)		
SGPT (U/L)	51.88 (43.11)		
GGT (U/L)	134.62 (213.14)		
Albumin (g/L)	43.12 (2.82)		
Total 25OH vitamin D level (nmol/L)	47.92 (38.06, 82.74)	38.69 (29.2, 52.42)	0.074
Immunogenic response and duration from first visit (days)			
Visit 1 (1st vaccination) (mean ± SD)	-	-	-
GMT of anti-RBD Ig (U/mL)	0.4	0.4	-
IFN-γ-secreting cells (SFCs/10^6^PBMCs)	8 (−1.197 to 17.197)	9.63 (4.25 to 15.01)	0.734
Visit 2 (2nd vaccination), days (mean ± SD)	21.3 (1.25)	28 (0)	<0.001
GMT of anti-RBD Ig (U/mL)	15.85 (5.25–47.87)	224.74 (126.60–398.93)	0.001
IFN-γ-secreting cells (SFCs/10^6^PBMCs)	86.59 (22.174 to 151.003)	182.37 (131.71 to 233.04)	0.019
Visit 3, days (mean ± SD)	34.93 (2.15)	42.07 (0.55)	<0.001
GMT of anti-RBD Ig (U/mL)	1511.37 (720.22–3171.59)	6311.90 (4955.46–8039.64)	0.009
IFN-γ-secreting cells (SFCs/10^6^PBMCs)	86.82 (−10.935 to 184.582)	365.04 (260.134 to 467.94)	<0.001
Visit 4, days (mean ± SD)	55.75 (1.87)	56.56 (1.80)	0.08
GMT of anti-RBD Ig (U/mL)	948.21 (550.05–1634.59)	4087.07 (3256.19–5129.97)	<0.001
IFN-γ-secreting cells (SFCs/10^6^PBMCs)	124.24 (26.105 to 222.366)	201.92 (227.293 to 291.84)	0.247
sVNT to Omicron variant (%)	8.23 (2.35, 16.73)	15.36 (4.61, 19.44)	0.066
positive	0/16	9/27	0.016
Anti-N IgG			
positive	0	0	-

Data presented as mean (95% confidence interval) unless otherwise noted. MMF: mycophenolate mofetil; Hb: hemoglobin; WBC: white blood cell; SGOT: serum glutamic oxaloacetic transaminase; SGPT: serum glutamate pyruvic transaminase; GGT: gamma-glutamyltransferase; IFN- γ: interferon gamma; RBD: the receptor-binding domain; sVNT: surrogate virus-neutralizing antibody; Anti-N: anti-nucleocapsid (anti-N); GMT: geometric mean titer.

**Table 2 vaccines-10-01324-t002:** Participant characteristics of liver-transplanted adolescents with low and high anti-RBD immunoglobulin levels after completion of the BNT162b2 vaccination.

Parameter	Liver-Transplanted Adolescents (*n* = 16)		*p* Value
	Anti-RBD IgG <550 U/L (*n* = 4)	Anti-RBD IgG ≥550 U/L	
		(*n* = 12)	
Age, years (mean ± SD)	15.30 (2.29)	14.34 (0.46)	0.368
Male gender, *n* (%)	1 (25)	6 (50)	0.585
Body mass index (kg/m^2^) (mean ± SD)	19.72 (2.71)	18.31 (2.52)	0.404
Time from LT to BNT162b2 vaccination (years) (mean ± SD)	7.99 (4.91)	7.73 (5.42)	0.931
History of graft rejection, *n* (%)	3 (75)	1 (8.33)	0.027
Type of Immunosuppressant, *n* (%)			
Tacrolimus (calcineurin inhibitor)	3 (75)	6 (50)	0.585
Cyclosporin (calcineurin inhibitor)	2 (50)	5 (41.67)	1
Sirolimus (mTor)	1 (25)	0	-
MMF (antimetabolite)	2 (50)	2 (16.67)	0.245
Prednisolone	3 (75)	0 (0)	0.007
Azathioprine	1 (25)	1 (8.33)	0.45
Immunosuppressant used (*n*), *n* (%)			0.003
1	0 (0)	9 (75)	0.019
2	1 (25)	3 (25)	1
3	3 (75)	0	0.007
Basic laboratory data (mean ± SD)			
Hb (g/L)	124.50 (21.98)	138.58 (22.23)	0.29
WBC (10^3^/µL)	5.52 (1.45)	5.34 (1.83)	0.858
Absolute neutrophil counts	3090.31 (1368.22)	2744.12 (767.00)	0.529
Absolute lymphocyte counts	1920.34 (550.74)	2030.43 (994.68)	0.838
Platelet (10^3^/microL)	220.50 (58.95)	198.42 (63.01)	0.548
Total bilirubin (µmol/L)	15.39 (6.84)	23.26 (20.18)	0.467
SGPT (U/L)	104.5 (46.34)	34.33 (24.61)	0.001
SGOT (U/L)	106.5 (65.40)	28.42 (16.70)	0.001
GGT (U/L)	276.75 (336.96)	87.25 (145.57)	0.127
Albumin (g/L)	42.75 (3.60)	43.25 (2.67)	0.769
Total 25OH vitamin D level (nmol/L)	42.31 (9.86)	60.85 (25.36)	0.134
sVNT to Omicron (%)	1.38 (1.82)	13.56 (10.00)	0.033

MMF: mycophenolate mofetil; Hb: hemoglobin; WBC: white blood cell; SGOT: serum glutamic oxaloacetic transaminase; SGPT: serum glutamate pyruvic transaminase; GGT: gamma-glutamyltransferase; sVNT: surrogate virus-neutralizing antibody.

**Table 3 vaccines-10-01324-t003:** Factors associated with sVNT to Omicron variant after BNT162b2 administration in multivariate regression analysis.

Factors	Difference *	95% CI	
Duration from LT to vaccination			
<3 years	Ref		
>3 years	7.875	0.745–15.005	0.03
Immunosuppression			
Monotherapy	Ref		
Combined therapy			
2 drugs	−8.503	−18.231–1.224	0.087
3 drugs	−14.373	−26.001–2.747	0.015

* Adjusted by potential confounders (age, sex and body mass index). LT: liver transplant; CI: confidence interval.

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
