# Peer review of "Safety and Humoral and Cellular Immunogenicity of the BNT162b2 SARS-CoV-2 Vaccine in Liver-Transplanted Adolescents Compared to Healthy Adolescents"

_vaccines, 2022, doi:10.3390/vaccines10081324_

Round 1

Reviewer 1 Report

The manuscript entitled “safety, humoral and cellular immunogenicity of the BNT162b2 SARS-CoV-2 vaccine in the liver transplanted adolescents comparing to healthy adolescents” by Sintusek is very interesting and suitable for publication. The following minor comments the author needs to address.

Why author used covid-19 vaccine for liver transplanted adolescents

What encourages to study the effect of COVID-19 vaccine potential in liver transplanted adolescents

The author needs to discuss the other applications for the covid-19 vaccine except for covid-19.

The author should discuss the novelty of the present study in the introduction

The number/values at the x-axis and y-axis of figures 1, 2, and 3 are too small and need to increase font size

Author needs to cite the literature https://www.sciencedirect.com/science/article/pii/S0169409X21000028

Author Response

10th August, 2022

Dear reviewers,

We thank the reviewers for the comments to our manuscript entitled “Safety, humoral and cellular immunogenicity of the BNT162b2 SARS-CoV-2 vaccine in liver-transplanted adolescents comparing to healthy adolescents”. The manuscript has been thoroughly revised according to the reviewers’ comments. We look forward to hearing your positive response to this revised manuscript. Below are the answers to the reviewer’s and editor’s comments.

Response to Reviewer 1

Comment: 1. Why author used covid-19 vaccine for liver transplanted adolescents

Answer: Thank you very much for reviewer’s questions. The reason we used BNT162b2 in liver transplanted adolescents was mentioned in the introduction part (line 47-51).

Comment: 2. What encourages to study the effect of covid-19 vaccine potential in liver transplanted adolescents

Answer: Thank you very much for reviewer’s questions. Because of our country’s policy to provide BNT162b2 that was approved by FDA, to all immunocompromised and healthy adolescents. However, there has been the limited data in aspect of the efficacy and safety of BNT162b2 in these patients. This study aims to evaluate these aspects (line 38-44).

Comment: 3. The author needs to discuss the other applications for the covid-19 vaccine except for covid-19.

Answer: We sincerely appreciate the reviewer’s helpful comments. We added more applications for the covid-19 including pre-exposure prophylaxis and treatment in the discussion part (line 355-360).

Comment: 4. The author should discuss the novelty of the present study in the introduction

Answer: Thank you very much indeed for your comment. We have added “To date, there has been few data of humoral response to BNT162b2 and no data of cellular response to BNT162b2  in liver transplanted children.” in the introduction part already (line 52-54).

Comment: 5. The number/values at the x-axis and y-axis of figure 1, 2, and 3 are too small and need to increase font size

Answer: Thank you very much indeed for your helpful comments. We have edited the number/values at the x-axis and y-axis of figure 1-3 as suggestion already.

Comment: 6. Author needs to cite the literature http://www.sciencedirect.com/science/article/pii/s0169409x21000028

Answer: Thank you very much for reviewer’s suggestion. We have cited this literature in the discussion part (line 356).

Reviewer 2 Report

In this manuscript, the authors evaluated the SARS-CoV-2 mRNA vaccine BNT162b2 in a specific group of individuals, the adolescents who underwent liver-transplantation. Given the vulnerability of this population, the authors investigated the vaccine immunogenicity inducing both humoral and cellular responses, as well as the local and systemic adverse effects, in the liver-transplanted adolescents comparing with healthy adolescents. Although the participant number is relatively low and the duration of the follow-up study is limited, this study provided important information regarding the safety and immunogenicity of the vaccine in these vulnerable individuals, with selection criteria carefully described, and potential factors influencing immunologic responses discussed. I would suggest this manuscript to be considered publication if below major and minor points are addressed by the authors:

Major points:

For vaccine immunogenicity for inducing humoral responses, the authors utilized the commercially available kits for testing serum binding titers against SARS-CoV-2 spike RBD, which is a very common method in many other studies. But the authors here were not consistent in describing whether they are testing total immunoglobulin or only IgG: in line 81 and in the title of figure 1 it was written as “total immunoglobulins” or “Ig”, while in line 84 as well as in the y-axis of figure 1 it was written as “IgG”). Therefore, it is necessary for the authors to clarify this. In addition, if only IgG was tested, IgA titers may be considered by the authors as well.

A surrogate virus neutralization test (sVNT) was utilized by the authors in this manuscript to assess the neutralizing antibodies from the serum samples. However, this assay was only to quantify the neutralizing antibodies that function through directly preventing interaction between viral RBD and host ACE2 receptor. Many recent studies revealed other types of neutralizing antibodies, more specifically, those targeting S2 subunit, that disrupt the steps after the binding process. For example, neutralizing antibodies targeting fusion peptide prevent the protease cleavage at the S2’ site, while those targeting S2 stem helix prevent spike protein refolding which is necessary for membrane fusion. Therefore, the titers of these types of neutralizing antibodies in the serum samples cannot be assessed in the sVNT assay. A pseudovirus neutralization assay or other potential assays might be necessary to make the conclusion about neutralizing antibodies more solid.

For vaccine immunogenicity for inducing cellular responses, besides interferon-γ-secreting T cells evaluated in the manuscript, it would be better to investigate the TNF-α-secreting or IL-2-secreting CD4+ and CD8+ T cells as well, so that the cellular immunogenicity evaluation as claimed in this study would be more complete.

It may be necessary to mention the difference in the timing of 2nd vaccination between liver-transplanted adolescent and healthy adolescent group: the 2nd vaccinations of liver-transplanted group were at week3, while those of the healthy group were at wk4. The authors may need to add a sentence or two discussing whether this may affect the boosting effect.

Minor points:

In the introduction section, line 35-36, the infection and death case numbers due to SARS-CoV-2 will need to be updated, and reference 3 will need to be changed.

In line 100 “SFCs/106 PBMC”, the “6” should be superscripted.

In all the tables, the text might look clearer if they are aligned to the left, especially the most left column describing the parameters.

In Table 1 at the very last row, for the anti-N IgG positive number, the column of liver-transplanted group was left blank.

In Figure 3, the x-axis should be consistent with Figure 1. Instead of “week 0, 3, 5, 8”, it should be “week 0, 3-4, 5-6, 8”.

In line 191 there is a typo, “inhibition … with SGPT and SGPT” should be “… with SGOT and SGPT”.

In line 267, “seem to less effective” should be “seem to be less effective”.

In line 279, “NAFLD” was not given its full name.

In Figure 5, the “A” and “B” indicating the two separate panels are missing. Besides, the format of x-axis should be adjusted. It looks too crowded in the current format, thus is not clear for the readers to get the information quickly. The names of “LT adolescents” and “Healthy adolescents” can be shortened as “LT” and “Healthy”. And there can be extra gaps or dotted lines to separate the columns indicating each type of adverse effects.

Author Response

 10th August, 2022

Dear reviewers,

We thank the reviewers for the comments to our manuscript entitled “Safety, humoral and cellular immunogenicity of the BNT162b2 SARS-CoV-2 vaccine in liver-transplanted adolescents comparing to healthy adolescents”. The manuscript has been thoroughly revised according to the reviewers’ comments. We look forward to hearing your positive response to this revised manuscript. Below are the answers to the reviewer’s and editor’s comments.

Response to Reviewer 2

Major points:

Comments: 1. For vaccine immunogenicity for inducing humoral responses, the authors utilized the commercially available kits for testing serum binding titers against SARS-CoV-2 spike RBD, which is a very common method in many other studies. But the authors here were not consistent in describing whether they are testing total immunoglobulin or only IgG: in line 81 and in the title of figure 1 it was written as “total immunoglobulins” or “Ig”, while in line 84 as well as in the y-axis of figure 1 it was written as “IgG”). Therefore, it is necessary for the authors to clarify this. In addition, if only IgG was tested, IgA titers may be considered by the authors as well.

Answers : Thank you very much indeed for your helpful comments. In line 85 we changed “total immunoglobulins specific to…” to “…total immunoglobulins (Ig) specific to….”. In line 89 and the y-axis of figure 1, we changed IgG to Ig, already.

Comments: 2. A surrogate virus neutralization test (sVNT) was utilized by the authors in this manuscript to assess the neutralizing antibodies from the serum samples. However, this assay was only to quantify the neutralizing antibodies that function through directly preventing interaction between viral RBD and host ACE2 receptor. Many recent studies revealed other types of neutralizing antibodies, more specifically, those targeting S2 subunit, that disrupt the steps after the binding process. For example, neutralizing antibodies targeting fusion peptide prevent the protease cleavage at the S2’ site, while those targeting S2 stem helix prevent spike protein refolding which is necessary for membrane fusion. Therefore, the titers of these types of neutralizing antibodies in the serum samples cannot be assessed in the sVNT assay. A pseudovirus neutralization assay or other potential assays might be necessary to make the conclusion about neutralizing antibodies more solid.

Answer: Thank you very much indeed for editor’s comment. We totally agree with the reviewer’s comment. So we added this important issue in the discussion part (line 277-280).

Comment: 3. For vaccine immunogenicity for inducing cellular responses, besides interferon-γ-secreting T cells evaluated in the manuscript, it would be better to investigate the TNF-α-secreting or IL-2-secreting CD4+ and CD8+ T cells as well, so that the cellular immunogenicity evaluation as claimed in this study would be more complete.

Answer: Thank you very much for the reviewer’s suggestion. We totally agree with you to investigate more especially the TNF-α-secreting or IL-2-secreting CD4+ and CD8+ T cells after BNT162b2. Unfortunately, we did not have enough cells to perform TNF-alpha and IL2-secreting CD4+ and CD8+ T cells by intracellular cytokine staining or sorting cells. Therefore, we only performed the human IFN-gamma ELISpot assay, which detected Ag-specific cytokine-secreting cells and measures IFN-gamma responses of both CD4+ and CD8+ effector memory T cells together. In addition, the IFN-gamma cytokines are an important cytokine secreting cells as indicator for immune response after vaccine immunization. However, we decided to discuss more in this important point as the limitation of our study, in the discussion part (line 346-351).

Comment: 4. It may be necessary to mention the difference in the timing of 2nd vaccination between liver-transplanted adolescent and healthy adolescent group: the 2nd vaccinations of liver-transplanted group were at week3, while those of the healthy group were at wk4. The authors may need to add a sentence or two discussing whether this may affect the boosting effect.

Minor points: Thank you very much for the reviewer’s comment. We have added this crucial point in the limitation of our study already (line 348-353)

Comment: 5. In the introduction section, line 35-36, the infection and death case numbers due to SARS-CoV-2 will need to be updated, and reference 3 will need to be changed.

Answer: We sincerely appreciate the reviewer’s helpful comments. We updated the data to “More than 579 million people were infected with SARS-CoV-2, and unfortunatedly, more than 6.4 million dealths occurred.”, and the reference 3 was changed to “World Health Organization (2022 S. WHO Coronavirus (COVID-19) Dashboard; 2022 [Cited 2022 August 5] [Internet] Available from: http://covid19who.int/”

Comment: 6. In line 100 “SFCs/106 PBMC”, the “6” should be superscripted.

Answers: We sincerely appreciate the reviewer’s comments. We have superscripted 106 to 106, already.

Comment: 7. In all the tables, the text might look clearer if they are aligned to the left, especially the most left column describing the parameters.

Answers: Thank you for your kind suggestion. All texts in all tables were aligned to the left already.

Comment: 8. In Table 1 at the very last row, for the anti-N IgG positive number, the column of liver-transplanted group was left blank.

Answers: Thank you for your kind suggestion. We have added the number of participants who had anti-N IgG positive (=0) in the table already.

Comment: 9. In Figure 3, the x-axis should be consistent with Figure 1. Instead of “week 0, 3, 5, 8”, it should be “week 0, 3-4, 5-6, 8”.

Answers: Thank you very much for your eagle eyes. We have edited the figure as the suggestion already.

Comment: 10. In line 191 there is a typo, “inhibition … with SGPT and SGPT” should be “… with SGOT and SGPT”.

Answers: We really appreciate the reviewer’s comment and have changed SGPT to SGOT already (line 200).

Comment: 11. In line 267, “seem to less effective” should be “seem to be less effective”.

Answer: We have corrected the grammar as reviewer’s suggestion already (now, it is in line 287).

Comment: 12. In line 279, “NAFLD” was not given its full name.

Answer: We add “Nonalcoholic fatty liver disease (NAFLD)” in the context already (now, it is in line 300-301)

Comment: 13. In Figure 5, the “A” and “B” indicating the two separate panels are missing. Besides, the format of x-axis should be adjusted. It looks too crowded in the current format, thus is not clear for the readers to get the information quickly. The names of “LT adolescents” and “Healthy adolescents” can be shortened as “LT” and “Healthy”. And there can be extra gaps or dotted lines to separate the columns indicating each type of adverse effects.

Answer: We sincerely appreciate the editor’s helpful comments. We added label A and B for figure 5. Furthermore, we also changed “LT adolescents” and “Healthy adolescents” to “LT” and “Healthy” as suggestion.

Once again, thank you very much indeed for your comments and suggestions that really help us to improve the quality of our manuscript. We will look forward to hearing your decision.

Sincerely yours,

Prof. Yong Poovorawan, M.D.

Excellence Center of Clinical Virology, Department of Pediatrics, Bangkok, Thailand, Department of Pediatrics, Faculty of Medicine, King Chulalongkorn Memorial Hospital, Chulalongkorn University, Bangkok, Thailand, 10330

Round 2

Reviewer 2 Report

The authors have address the points I raised in the current version, therefore I recommend publication of this manuscript.